

# Impact of the coronavirus disease 2019 pandemic on the diversity of notifiable infectious diseases: a case study in Shanghai, China

Yongfang Zhang and Wenli Feng

School of Chemistry and Chemical Engineering, Zhoukou Normal University, Zhoukou, China

## ABSTRACT

The outbreak of coronavirus disease 2019 (COVID-19) has not only posed significant challenges to public health but has also impacted every aspect of society and the environment. In this study, we propose an index of notifiable disease outbreaks (NDOI) to assess the impact of COVID-19 on other notifiable diseases in Shanghai, China. Additionally, we identify the critical factors influencing these diseases using multivariate statistical analysis. We collected monthly data on 34 notifiable infectious diseases (NIDs) and corresponding environmental and socioeconomic factors (17 indicators) from January 2017 to December 2020. The results revealed that the total number of cases and NDOI of all notifiable diseases decreased by 47.1% and 52.6%, respectively, compared to the period before the COVID-19 pandemic. Moreover, the COVID-19 pandemic has led to improved air quality as well as impacted the social economy and human life. Redundancy analysis (RDA) showed that population mobility, particulate matter (PM2.5), atmospheric pressure, and temperature were the primary factors influencing the spread of notifiable diseases. The NDOI is beneficial in establishing an early warning system for infectious disease epidemics at different scales. Furthermore, our findings also provide insight into the response mechanisms of notifiable diseases influenced by social and environmental factors.

## INTRODUCTION

Rapid reporting systems for infectious diseases are important for swift public health action and provide fundamental information on disease epidemiology at local, regional, and national levels (*Brabazon et al., 2015*). Serving as a cornerstone of public health data, reports on notifiable infectious diseases (NIDs) can offer new early-warning signs for tackling infectious diseases and public health emergencies. The World Health Organization (WHO) advocated for worldwide voluntary reporting of selected infectious diseases. Numerous countries have established systems for reporting infectious diseases. In China, 40 infectious diseases, including coronavirus disease 2019 (COVID-19), were designated as notifiable in 2020, with monthly reports published on the public health

Corresponding author
Wenli Feng, wlfeng@zknu.edu.cn

department website. In the United States, the number of notifiable diseases varies from 35 to 120, depending on the local laws of each state (*Kirch, 2008*). Different infectious diseases react differently under specific spatiotemporal conditions and to various factors. The dynamics of notifiable diseases are influenced by various factors, such as meteorological factors, environmental conditions, and socioeconomic and demographic factors (*Liang et al., 2014*; *Travaglio et al., 2021*; *Sarkodie & Owusu, 2021a*). Therefore, it is important to study these influencing factors from multiple perspectives for effective prevention and control of infectious diseases.

The novel coronavirus disease outbreak, which began in December 2019, has been declared a public health emergency of international concern. The outbreak has tested public health systems worldwide in their efforts to prevent and control the spread of the epidemic. As of September 2021, there were nearly 230 million confirmed cases of COVID-19 globally, with approximately 4.7 million deaths (*World Health Organization, 2021*). During the COVID-19 pandemic, many countries and regions implemented a series of prevention measures, such as home quarantine, lockdowns, and restrictions on travel and non-essential public activities. For example, in Shanghai, within one week after the first case of COVID-19 was confirmed on January 22, 2020, the public gatherings were banned, a Level-1 response of Public Health Emergency was launched and then the shutdown of the workplace was prolonged. These measures led to reduced traffic emissions, industrial activities, and construction work in current report (*Sarkodie & Owusu, 2021b*). Thus, the COVID-19 pandemic not only influenced social development and public health, but also impacted the environment. In China, the total retail sales of consumer goods decreased by 16.2% during the first two months of 2020, the critical period of the pandemic, compared with the same period in the previous year. The total import and export value dropped by −11%, and passenger traffic volume was reduced by 87.2%, affected by restrictions on population flow. Although the socioeconomic loss has not been fully accounted for in the traditional gross domestic product (GDP) calculations, it merits attention in future economic studies (*Ali & Alharbi, 2020*; *Liu, Zheng & Zhang, 2021*). The International Monetary Fund (IMF) forecasts at least a 3% contraction in the global economy, due to uncertainty and the reduction in the movement of goods and population occurring as a result of caused by the pandemic (*International Monetary Fund, 2020*). There is no doubt that the emerging COVID-19 pandemic imposes a substantial social and economic burden worldwide (*Kim et al., 2021*). However, the impact of changes in social development and economic factors due to the COVID-19 pandemic on NIDs and public health is still not fully understood.

Recent peer-reviewed studies have reported that the environmental indicators including air and surface water pollutants, the volume of municipal solid waste and greenhouse gas emissions. As study on 44 cities in northern China showed that the concentrations of $SO_2$, $PM_{2.5}$, $PM_{10}$, $NO_2$, and CO decreased by 6.76%, 5.93%, 13.66%, 24.67%, and 4.58%, respectively, along with a drop of 69.85% in population mobility during the lockdown. Moreover, $NO_2$ pollution in New York (USA) was reduced by 30%, with same-period reductions of 40–50% in cities across Asia and Europe during the lockdown (*Tobias et al., 2020*). On the contrary, the increased concentrations of $O_3$ in many regions were observed

in many regions during the lockdown (*Sicard et al., 2020*; *Suhaimi, Jalaludin & Latif, 2020*; *Ju, Oh & Choi, 2021*). Interestingly, surface $O_3$ concentrations tend to be higher on weekends compared to weekdays, indicating a clear "weekend effect", despite lower emissions of NOx, VOCs and PMs (*Sicard et al., 2020*). Previous studies have explained and helped elucidate the changing mechanisms of $O_3$ formation (*Huryn & Gough, 2014*; *Feng et al., 2021*). Many studies have shown that some air pollutants, such as $PM_{2.5}$ and $PM_{10}$, can carry microorganisms to the respiratory system, affecting immunity and making individuals more susceptible to pathogens (*Ju, Oh & Choi, 2021*; *Travaglio et al., 2021*). Additionally, several early studies have demonstrated that population mobility has a significant influence on the COVID-19 epidemic and other infectious diseases (*Aral & Bakir, 2022*; *Kolluru et al., 2021*). Additionally, several early studies have demonstrated that population mobility has a significant effect on the COVID-19 epidemic and other infectious diseases (*Maltezou et al., 2013*; *Kraemer et al., 2020*; *Yang, 2020*). Currently, we already understand the impact of the COVID-19 epidemic on these factors. However, the question arises: How do these factors influence the spread of various other infectious diseases? Therefore, it is intriguing and worthwhile to study the response mechanism of other NIDs to changes in environmental and socioeconomic factors caused by the COVID-19 pandemic.

The objectives of this study are to (a) investigate the impact of the COVID-19 pandemic on environmental and socioeconomic factors, (b) determine the influence of COVID-19 on notifiable diseases and the index of notifiable disease outbreaks (NDOI), and (c) explore the critical factors influencing NIDs. This study will provide a new method to assess the influences of various factors on NIDs. The findings will help us understand the relationship between infectious disease epidemics and environmental and socioeconomic factors, and optimize intervention measures to control the spread of infectious diseases during pandemics.

## MATERIALS AND METHODS

### Data sources

To estimate the influence of the COVID-19 pandemic on other notifiable diseases in Shanghai, China, monthly data on air pollution, meteorological factors, and socio-economic factors were collected from January 1, 2017 to December 31, 2020. Data on air-quality index (AQI), particulate matter 2.5 ($PM_{2.5}$, μg/m$^3$), particulate matter 10 ($PM_{10}$, μg/m$^3$), sulfur dioxide ($SO_2$, μg/m$^3$) nitrogen dioxide ($NO_2$, μg/m$^3$), carbon monoxide (CO, mg/m$^3$) and ozone ($O_3$, μg/m$^3$), were extracted from the China Air Quality online monitoring and analysis platform. Meteorological data, including atmospheric pressure (kPa), ambient temperature (°C), relative humidity (%), wind speed (m/s), and precipitation were obtained from NASA research supporting renewable energy, building energy efficiency and agricultural needs (*National Aeronautics and Space Administration, 2020*). Socioeconomic data were collected from monthly reports released on the official website of the *Shanghai Bureau of Statistics (2020)*. In this study, some indicators related to socioeconomic development including human flow, logistics flow, social consumption and industrial output were also considered. Specifically, they included inbound travel, airport

passenger throughput, freight volume, and gross industrial output value, all of which are released monthly on the report. The monthly cases of 40 NIDs were obtained from reports released on the website of the *Shanghai Municipal Health Commission (2020)*.

### Index of notifiable disease outbreaks

In this study, we proposed an index of notifiable disease outbreaks (NDOI) to assess the relative intensity of notifiable disease outbreaks different areas or periods:

$$NDOI_i = \alpha \sum_{j=1}^{n} N_{i,j} + \beta \sum_{j=1}^{t} T_{i,j} \tag{1}$$

where $NDOI_i$ represents the index of outbreaks in a period of time $i$, which can be used to calculate the NDOI at a specific time or for higher categories of various diseases, such as transmission route $j$. $\alpha$ and $\beta$ are weight coefficients for the normalized number of confirmed cases ($N_i$) and normalized types of disease ($T_i$).

$$N_i = \frac{n_i - \min(n_i)}{\max(n_i) - \min(n_i)} \tag{2}$$

$$T_i = \frac{t_i - \min(t_i)}{\max(t_i) - \min(t_i)} \tag{3}$$

where $n_i$, and $t_i$ are the number of confirmed cases and types of disease, respectively. According to the calculation of the entropy weight method, $\alpha = 0.51$, $\beta = 0.49$ (Section S1 in Supporting Information and Table S1), the number of monthly confirmed cases and types of diseases has equal weight for each sample, therefore, the NDOI is a normalized with a range of 0 to 1. The maximum values of these two variables can be obtained from the historical reports, and the minimum values may be observed under conditions of effective epidemic control.

### Statistical analysis

The Kolmogorov-Smirnov test was used to test the normality of all data, and data that did not follow a normal distribution were log-transformed (*Feng et al., 2021*). To evaluate the impact of the COVID-19 pandemic on notifiable diseases and variables, the number of confirmed cases, air pollutants, and socioeconomic variables were compared before and during the COVID-19 pandemic. One-way ANOVA was used to identify differences among groups of variables, with the significance level set at $p$-value $< 0.05$.

For the forecasting of monthly NIDs cases, a predictor on the Oracle Crystal Ball (version 11.1) was employed to predict the tendency of confirmed cases, types and NDOI during the COVID-2019 epidemic. The forecasting performance was assessed using error metrics including mean absolute deviation (MAD), mean absolute percentage error (MAPE), and root mean square error (RMSE) (*Punyapornwithaya et al., 2023*). To investigate the relationships among notifiable diseases, various variables, and samples, we used multivariate statistical analyses, including principal component analysis (PCA), detrended correspondence analysis (DCA) and redundancy analysis (RDA). This approach was chosen because the gradient length of the first axis of DCA was 0.5.

The statistical significance of the environmental variables was assessed using a Monte Carlo test with 499 permutations. Prior to the RDA analysis, rare species (some infectious diseases) with a frequency lower than 5% were omitted. Finally, 22 notifiable diseases were incorporated into the multivariate statistical analysis. The Kolmogorov-Smirnov test and one-way ANOVA tests were carried out using SPSS statistical software (version 25; IBM, Armonk, NY, USA). Multivariate statistical analysis was performed using the CANOCO software package (version 5.0; Biometris, Ladysmith, VA, USA) (*Feng et al., 2019*).

## RESULTS AND DISCUSSION

### Influence of COVID-19 on environmental and socioeconomic factors

Figure 1 shows the changes in monthly meteorological factors, air pollutants, AQI and socioeconomic factors influenced by the COVID-19 pandemic. The monthly variations of these factors from 2017 to 2020 in Shanghai are shown in Fig. S1. The results showed that there was no significant difference between the average monthly values in 3 years, from 2017 to 2019 (before the COVID-19 outbreak) and those during the COVID-19 pandemic (2020) (Fig. 1A). Therefore, the COVID-19 pandemic did not significantly affect meteorological factors. Most factors, except for wind speed, showed strong regular seasonal changes, as shown in Fig. S1A. However, the COVID-19 pandemic led to a reduction in air pollution levels. During the COVID-19 pandemic, the concentrations of $PM_{10}$ and $SO_2$ decreased significantly by 18.0% and 30.4%, respectively. In addition, $PM_{2.5}$, $NO_2$, and AQI also decreased by 11.5%, 10.6%, and 10.0%, respectively. These findings suggest that the COVID-19 pandemic contributed to a reduction in pollutant concentrations and an improvement in air quality in Shanghai, China. The COVID-19 outbreak also influenced socioeconomic development. The number of monthly inbound travelers dropped by 85.5% compared to the period before the COVID-19 pandemic. Similarly, airport passenger throughput decreased by 47.4%. However, freight volume and total retail sales of consumer goods increased by 32.9% and 25.8%, respectively, possibly due to tight supplies and rising demand for social consumer goods.

Given the complexity of the emergency, governments in many countries implemented a series of policies and measures, such as national lockdown policies to restrict travel, shutdown of commercial activities, and stay-at-home orders (*Zhang et al., 2020*; *Niu et al., 2021*). These measures significantly influenced the atmospheric environment, drawing widespread attention from scholars. Indeed, many studies have reported that the COVID-19 pandemic has played an important role in improving ambient air quality (*Bao & Zhang, 2020*; *Mahato, Pal & Ghosh, 2020*; *Sicard et al., 2020*; *Stratoulias & Nuthammachot, 2020*; *Ju, Oh & Choi, 2021*; *Wang et al., 2021*; *Zoran et al., 2022*). Thus, we also analyzed the changes in air pollutants triggered by the COVID-19 pandemic in several metropolises with varying meteorological conditions in China. As shown in Table 1, most air pollutants have been reduced by the COVID-19 pandemic in many cities or countries. For example, the concentrations of $PM_{2.5}$ and $PM_{10}$ were decreased by 3–36% and 4.71–37.71%, respectively. Additionally, the concentrations of $NO_2$, $SO_2$, and CO were reduced by 1.31–52.68%, 2.38–20.45%, and 1–100%, respectively. The concentrations of all monitored pollutants were decreased in metropolitan regions across China, such as Beijing,

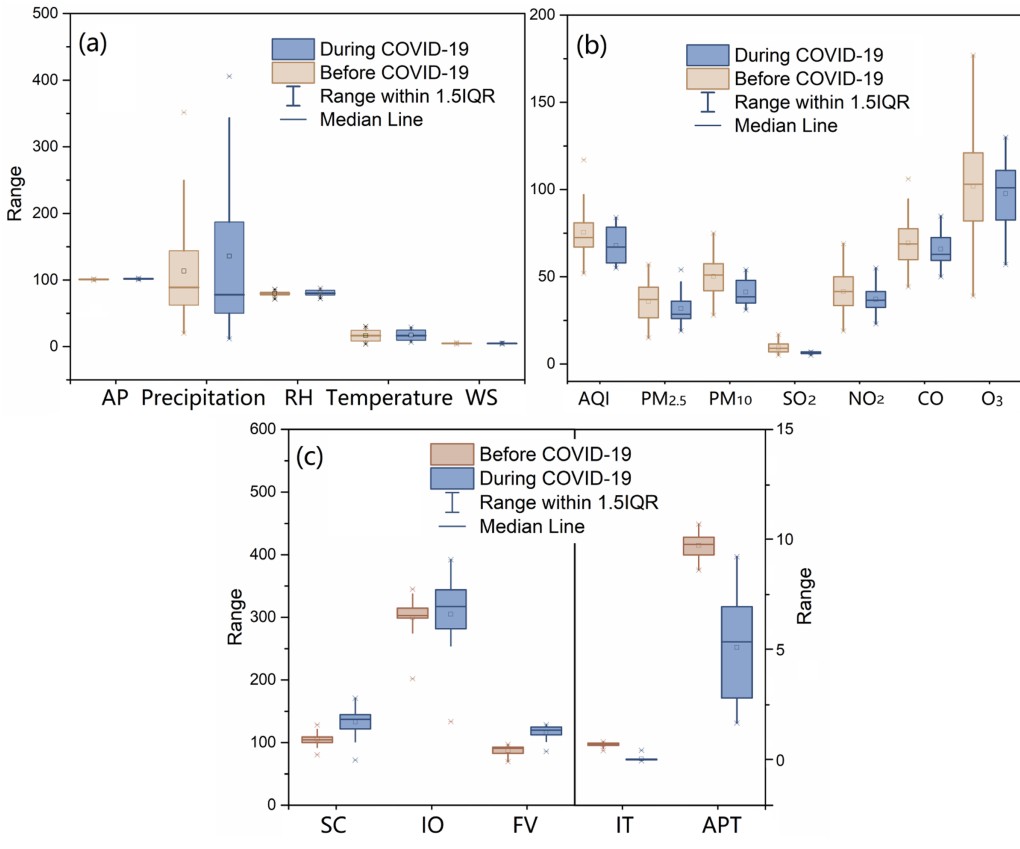

**Figure 1 Changes in environmental and socioeconomic variations of Shanghai impacted by COVID-19 pandemic.** (A) The comparison of meteorological factors between before and during COVID-19 pandemic; AP, atmosphere pressure; kPa, precipitation, mm; d; RH, relative humidity, %; temperature, °C; WS, wind speed, m s$^{-1}$; (B) the comparison of atmospheric environmental factors between before and during COVID-19 pandemic; AQI, air quality index; (C) the comparison of socioeconomic factors between before and during COVID-19 pandemic; SC, Social consumption, billion yuan; IO, industrial output value, billion yuan; FV, Freight volume, million ton; IT, Inbound travel, million travelers; APT, airport passenger throughput, million passengers.

Zhengzhou, and Shanghai. Many recent studies have highlighted that anthropogenic activities such as industrial production, traffic, and transportation are major sources of air pollutants (*Bao & Zhang, 2020*; *Prakash et al., 2021*; *Sathe et al., 2021*; *Wang et al., 2021*). Thus, the implementation of full or partial lockdowns during the COVID-19 pandemic, resulting in reduced human activities, contributed significantly to alleviating severe air pollution in Shanghai. In contrast, concentrations of O$_3$ increased in cities like Barcelona, Nice, and Rome due to measures taken during the COVID-19 pandemic (*Sicard et al., 2020*; *Tobias et al., 2020*). The increase in O$_3$ concentrations may be attributed to an unprecedented reduction in NOx emissions, leading to lower O$_3$ titration by NO, and a decrease in PMs, which in turn caused higher-levels of solar radiation (*Sicard et al., 2020*). This phenomenon provides insight into the effectiveness of policies for the reduction of emissions of O$_3$ precursor pollutants. Overall, the findings indicate that measures carried
**Table 1 Reduction of in air pollutants in some countries, cities, and region during COVID-19 pandemic (the plus signs denote the value is percentage increase to before COVID-19).**

| | $PM_{2.5}$ | $PM_{10}$ | $NO_2$ | $SO_2$ | CO | $O_3$ | Sources |
|---|---|---|---|---|---|---|---|
| Shanghai, China | 11.46% | 17.96% | 30.42% | 10.64% | 5.00% | 4.09% | This study |
| Beijing, China | 9.72% | 15.80% | 15.09% | 20.45% | 10.04% | 1.81% | This study |
| Zhengzhou, China | 13.07% | 14.74% | 8.04% | 10.18% | 13.81% | 2.56% | This study |
| Chongqing, China | 13.66% | 9.16% | 1.31% | 5.32% | +2.22% | 8.34% | This study |
| Northern area, China | 5.93% | 13.66% | 24.67% | 6.76% | 4.58% | NA | *Bao & Zhang (2020)* |
| Harbin, China | +12.18% | 8.41% | 2.05% | +6.06% | 2.75% | +9.78% | This study |
| Nanjing, China | 22.31% | 21.81% | 14.37% | 3.95% | 29.41% | 2.91% | This study |
| Lanzhou, China | 4.05% | 4.71% | 4.88% | 2.38% | 14.15% | +2.33% | This study |
| Guangzhou, China | 22.84% | 18.97% | +2.47% | 9.40% | 18.44% | 2.80% | This study |
| USA | 16.48% | 1.88% | 34.41% | – | 45.45% | 23.63% | *Bekbulat et al. (2021)* |
| Delhi, India | 60% | 39% | 52.68% | NA | 30.35% | +0.78% | *Mahato, Pal & Ghosh (2020)* |
| Korea | 45% | 36% | 20% | NA | 17% | NA | *Ju, Oh & Choi (2021)* |
| Kuala Lumpur, Malaysia | 3–36% | 28% | 43–68% | 6–48% | 1–48% | +21% | *Suhaimi, Jalaludin & Latif (2020)* |
| Barcelona, Spain | NA | 27.8–31.0% | 47.0–51.4% | 19.4–+1.8% | NA | +28.5–+57.7% | *Tobias et al. (2020)* |
| Hat Yai, Thailand | 21.8% | 22.9% | 33.7 | NA | 9.9 | 12.5 | *Stratoulias & Nuthammachot (2020)* |
| Sao Paulo, Rio De Janeiro, Brazil | NA | +20.7–37.7% | 14.5–41.8% | NA | 30–100% | NA | *Siciliano et al. (2020)* |
| Roman, Itlay | +22.6% | +18.5% | 45.6% | NA | NA | +14% | *Sicard et al. (2020)* |
| Nice, France | 19.0% | 7.8% | 62.8% | NA | NA | +24–+27% | *Sicard et al. (2020)* |

out during the COVID-19 pandemic actually reduced the levels of most pollutants and improved air quality in many regions and countries.

Some socioeconomic indicators were also influenced by the COVID-19 pandemic. In this study, we observed a significant reduction in the number of monthly inbound travelers and airport passenger throughput. During the COVID-19 outbreak in late January 2020, Shanghai implemented several stringent containment measures such as school closure, public transport closure, restrictions on gathering, stay-at-home requirements, and international travel controls (*Zhang et al., 2020*; *Niu et al., 2021*). In contrast, freight volume and total retail sales of consumer goods increased, possibly due to tight supplies and the rising demand for social consumer goods. In February 2020, consumer prices in Shanghai rose by 3.0% year-on-year (*Shanghai Bureau of Statistics, 2020*). The socioeconomic response to alleviate the economic burden included income support, fiscal measures, and international support (*Sarkodie & Owusu, 2021a*, *2021b*). In order to reduce the influence of the COVID-19 pandemic on economic development, by February 20, 2020, the return-to-work ratios for large businesses and industrial enterprises above a designated size had exceeded 70% (*Shanghai Bureau of Statistics, 2020*). We selected these socioeconomic development indicators to represent human flow, logistics flow, social consumption, and industrial output in Shanghai and to fully explore potential socioeconomic factors.

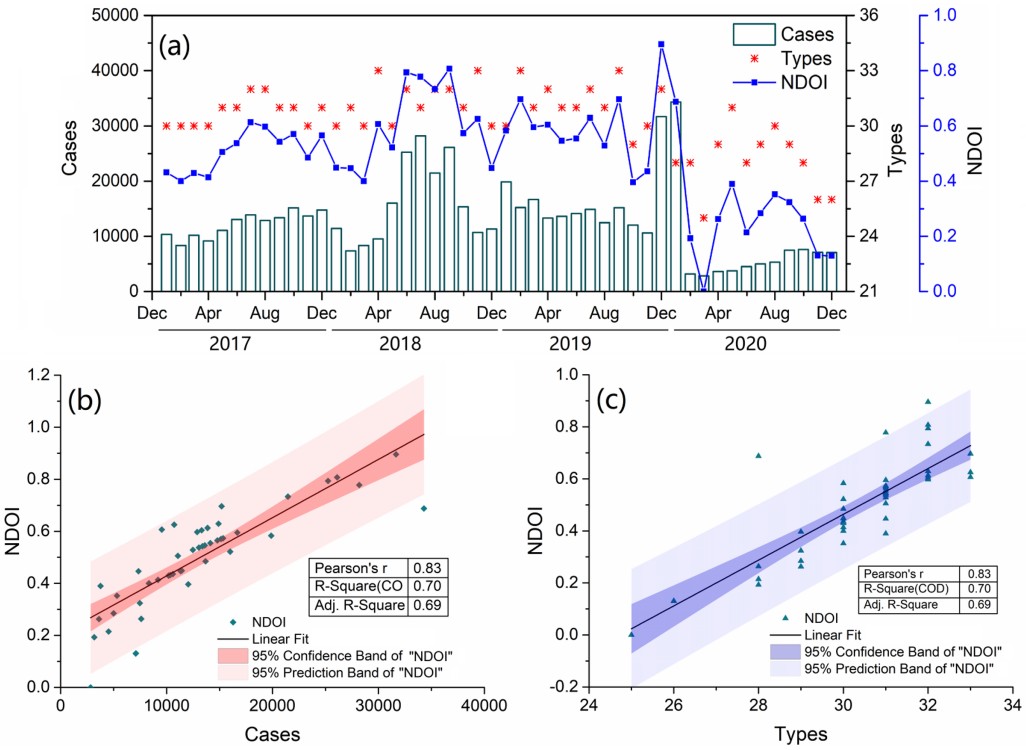

**Figure 2** (A) Change in the number of monthly confirmed case, type and index of notifiable disease outbreaks (NDOI) of 34 notifiable infectious diseases from 2017 to 2020. Pearson correlation analysis was employed to reveal the relationship between NDOI and the number of confirmed case (B) and type (C).

## Influence of COVID-19 on notifiable diseases

Overall, 32 notifiable diseases with four main transmission routes were included in this study. As shown in Fig. 2, there were a total of 145,986, 191,084 and 189,752 confirmed cases of notifiable diseases in 2017, 2018 and 2019, respectively, with an average of 175,607 cases. However, the number of confirmed cases in 2020 was 92,955, excluding 1,072 cases related to COVID-19. Therefore, during the COVID-19 pandemic, the number of confirmed cases decreased by 47.1%, and the types of notifiable diseases dropped from 31 to 28. For the NDOI, the monthly averages in 2017, 2018 and 2019 were 0.508, 0.599, and 0.597, respectively, but only 0.269 in 2020, a decrease of 52.6%. There was a significant correlation between the NDOI and the number or types of disease cases (Figs. 2B and 2C). Therefore, using the NDOI as a comprehensive index to reflect changes in various notifiable diseases is reasonable.

During the COVID-19 pandemic, changes in the number of notifiable disease cases varied according to their types and routes of transmission. As shown in Table 2, there were nine diseases with direct-contact transmission, with an annual average number of cases of 51,957 from 2017 to 2019. However, in 2020 the average number of cases was 18,319, a decrease of 64.7%. Meanwhile, the NDOI dropped by 48.5% by 2020. Among these, the most frequently occurring diseases were hand-foot-and-mouth disease (HMFD), syphilis, gonorrhea, acquired immune deficiency syndrome (AIDS), and measles, with an annual

**Table 2  Summary of confirmed cases, rate of change and index of notifiable disease outbreaks (NDOI) by four transmission routes during the study period.**

| Notifiable infectious diseases | Confirmed cases | | | | | Rate of change |
|---|---|---|---|---|---|---|
| | 2017 | 2018 | 2019 | Average | 2020 | |
| **Direct-contact transmitted diseases** | | | | | | |
| **NDOI** | **0.384** | **0.557** | **0.344** | **0.428** | **0.221** | **−48.5%** |
| **Cases** | **44,671** | **69,950** | **41,251** | **51,957** | **18,319** | **−64.7%** |
| Hand-foot-and-mouth disease (HFMD) | 24,292 | 52,976 | 247,88 | 34,019 | 5,619 | −83.5% |
| Syphilis | 13,932 | 12,447 | 12,637 | 13,005 | 9,721 | −25.3% |
| Gonorrhoea | 5836 | 3908 | 3188 | 4311 | 2,540 | −41.1% |
| Acquired immune deficiency syndrome (AIDS) | 555 | 546 | 562 | 554 | 423 | −23.7% |
| Measles | 49 | 57 | 71 | 59 | 10 | −83.1% |
| Brucellosis | 4 | 8 | 2 | 5 | 4 | −14.3% |
| Leprosy | 1 | 2 | 2 | 2 | 2 | 20.0% |
| Rabies | 2 | 5 | 0 | 2 | 0 | −100.0% |
| Hydatid disease (HD) | 0 | 1 | 1 | 1 | 0 | −100.0% |
| **Water and food transmitted diseases** | | | | | | |
| **NDOI** | **0.578** | **0.458** | **0.485** | **0.507** | **0.231** | **−54.4%** |
| **Cases** | **9,038** | **7,855** | **7,840** | **8,244** | **3,858** | **−53.2%** |
| Infectious diarrhea (ID) | 6,627 | 5,661 | 5,961 | 6,083 | 2,804 | −53.9% |
| Hepatitis A | 302 | 272 | 401 | 325 | 224 | −31.1% |
| Hepatitis B | 919 | 970 | 940 | 943 | 542 | −42.5% |
| Hepatitis C | 615 | 479 | 243 | 446 | 172 | −61.4% |
| Hepatitis D | 336 | 325 | 150 | 270 | 39 | −85.6% |
| Hepatitis E | 96 | 74 | 75 | 82 | 32 | −60.8% |
| Hepatitis (unspecified) | 94 | 44 | 35 | 58 | 20 | −65.3% |
| Bacillary and amoebic dysentery (BAD) | 0 | 2 | 1 | 1 | 1 | 0.0% |
| Typhoid and paratyphoid (TP) | 30 | 15 | 18 | 21 | 16 | −23.8% |
| Acute hemorrhagic conjunctivitis (AHC) | 19 | 13 | 16 | 16 | 8 | −50.0% |
| **Airborne transmitted diseases** | | | | | | |
| **NDOI** | **0.255** | **0.313** | **0.434** | **0.334** | **0.174** | **−47.9%** |
| **Cases** | **19,120** | **17,568** | **45,583** | **27,424** | **24,701** | **−9.9%** |
| Influenza | 6,144 | 4,201 | 33,115 | 14,487 | 15,706 | 8.4% |
| Pulmonary tuberculosis (PT) | 6,435 | 6,516 | 6,223 | 6,391 | 5,869 | −8.2% |
| Scarlatina | 4,234 | 4,423 | 3,726 | 4,128 | 812 | −80.3% |
| Mumps | 2,209 | 2,290 | 2,098 | 2,199 | 1,216 | −44.7% |
| Pertussis | 84 | 112 | 149 | 115 | 9 | −92.2% |
| Rubella | 4 | 21 | 265 | 97 | 15 | −84.5% |
| Epidemic hemorrhagic fever (EHF) | 5 | 2 | 2 | 3 | 2 | −33.3% |
| Epidemic cerebrospinal meningitis (ECM) | 1 | 3 | 5 | 3 | 0 | −100.0% |
| Human infection with $H_7N_9$ bird flu ($H_7N_9$) | 4 | 0 | 0 | 1 | 0 | −100.0% |
| **Vector-borne transmitted diseases** | | | | | | |
| **NDOI** | **0.413** | **0.399** | **0.530** | **0.447** | **0.268** | **−40.0%** |

| Table 2 (continued) | | | | | | |
|---|---|---|---|---|---|---|
| Notifiable infectious diseases | Confirmed cases | | | | | Rate of change |
| | 2017 | 2018 | 2019 | Average | 2020 | |
| **Cases** | 63 | 63 | 132 | 86 | 31 | −64.0% |
| Malaria | 42 | 26 | 25 | 31 | 18 | −41.9% |
| Dengue | 20 | 36 | 106 | 54 | 11 | −79.6% |
| Epidemic encephalitis B (EEB) | 1 | 1 | 1 | 1 | 1 | 0.0% |
| Kala-azar | 0 | 0 | 0 | 0 | 1 | NA |

**Note:**
The parameters in bold under the categories of infectious diseases are confirmed cases and NDOI which can be calculated from Eq. (1).

average number of cases that decreased by 23.7–83.5% compared to the 3-year average before the COVID-19 pandemic. Ten diseases transmitted through water and food, saw a 53.2% decrease in cases, from an average of 8,244 in 2017–2019 to 3,858 in 2020, with the NDOI decreasing by 54.4% in 2020. Of the seven frequent clinic diseases, including infectious diarrhea and six types of hepatitis, cases decreased by 31.1–85.6% during the COVID-19 pandemic compared to the average before 2020. Nine diseases transmitted by air or droplets had an annual number of 23,629 cases in 2020, a 13.8% decrease compared to before COVID-19, with the NDOI reducing by 47.9% in 2020. Six of these diseases, with a higher incidence than EHF, ECM, and human infection with H7N9 bird flu, saw an annual average case decrease of 8.2–84.5% from 2017–2019 to 2020. For the four vector-borne diseases, the number of cases decreased by 64.0%, with the NDOI dropping by 40.0% in 2020. Malaria and dengue, parasitic and viral diseases, respectively, mostly found in tropical and sub-tropical regions, saw an annual average case decrease of 41.9% and 79.6%, respectively, compared to the average of 2017–2019. There were no cases of rabies, HD, ECM, or H7N9 bird flu in 2020. Comparing the forecast results to the observed results during the COVID-2019 epidemic clarifies the understanding of the influence of COVID-19 on notifiable diseases. Figure S2 shows the monthly number of cases that decreased by 48.0–76.6% compared to the forecast number, with the NDOI decreasing by 32.7–100.0%. The Ljung-boxtest results were $P > 0.05$, and the residual sequence could be considered as white noise sequence (Table S2). Therefore, the selected model is effective and can be used for predicting time series.

COVID-19, a respiratory disease with various transmission routes, including direct contact, droplets and airborne transmission, prompted the advocacy of many protective measures during the pandemic, such as wearing masks, social distancing, frequent hand washing, ensuring adequate ventilation, avoiding crowded public places, and engaging in more exercise (*Ali & Alharbi, 2020*; *Bai, Jiang & Hou, 2022*; *Lai et al., 2021*). These practices likely reduced the risk of contracting other notifiable diseases. Most respiratory infections are transmitted by the airborne route through the inhalation of droplet nuclei, which from 0.5 to 5.0 µm diameter (*Cole & Cook, 1998*). The particle filtration efficiency (PFE) of face mask should exceed 95% meets the demands of technical standard (*Technical requirements for protective face mask for medical use, GB19083-2010*). The sodium chloride

aerosol with particle size of 0.24 ± 0.06 μm as filter object was conducted in filtration test. According to market survey report, in the product quality sampling percent of pass is 93.9% in Shanghai (*Shanghai Market Supervision Administration, 2020*). The horizontal distance of droplet spread depends on various factors, such as expiration fluid, ventilation, droplet, and expiration, *etc.*, the majority of droplets were expelled is 1 m. The 1 to 2 m of distance rule recommended by some guidelines comes epidemiologic and simulated studies. But more recent studies have shown the extent of droplet spread to be greater than 2 m (*Bahl et al., 2022*). To a certain extent, social distancing may decrease the population risk of exposure to NIDs, for indirect transmission is more significant, the effectiveness of social distancing would be reduced (*Sajjadi, Hashemi & Ghanbarnejad, 2020*). Washing hands is one of the simplest and most effective measures to prevent infectious diseases. According to a quantitative systematic, promoting the frequency handwashing lowered risks of respiratory infection, with risk reductions ranging from 6% to 44% and reduced incidence of diarrhea by 41% (*Rabie & Curtis, 2006*; *Ataee et al., 2017*). Historical and recent studies suggest that natural ventilation offers protection from transmission of airborne pathogens. Fresh air can provide additional microbiocidal and/or attenuating effects other than physical dilution or displacement of airborne pathogens. It is worth noting that infection risk is lowered when personalized ventilation delivers clean air to inhalation without direct entrainment from infectious exhalation (*Hobday & Dancer, 2013*). Physical exercise as an auxiliary tool in strengthening and preparing the immune system is benefit for the response to viral communicable diseases (*Simpson & Katsanis, 2020*). The effect of single strategy is typically limited, and the combining multiple interventions, in general, is effective to prevent the infectious disease pandemic (*Oduro & Magagula, 2021*).

In this study, a decrease of 52.6% in notifiable disease cases was observed in Shanghai in 2020. Similar findings have been reported in other studies; for example, these protective measures led to a 25–100% reduction in the number of diseases cases transmitted by four different routes in 2020 compared to 2019 in Taiwan (*Lai et al., 2021*). In addition, respiratory infectious diseases in China decreased by 60–90%, and diseases transmitted *via* the digestive tract and animal-borne diseases saw reductions of about 20–30% in 2020 (*Bai, Jiang & Hou, 2022*). Similar results were obtained in the USA, Italy, Japan, Finland and Singapore (*Chow, Hein & Kyaw, 2020*; *Sakamoto, Ishikane & Ueda, 2020*; *Kuitunen & Ponkilainen, 2021*), supporting the notion that prevention and control measures may have a similar effect in preventing infections of notifiable diseases. Furthermore, changes in environmental and socioeconomic factors under these measures during the COVID-19 pandemic are likely responsible for the reduction in cases.

## Potential influencing factors

Redundancy analysis (RDA) was performed to determine the influence of environmental variables and socioeconomic factors on notifiable diseases. As shown in Fig. 3 and Table S3, the first two axis of the RDA analysis explained 69.7% of the total variance, with Axis 1 explaining 48.7% of the variance and Axis 2 accounting for 21.0%, in relation to the composition of notifiable diseases. The results highlight the influence of various factors on

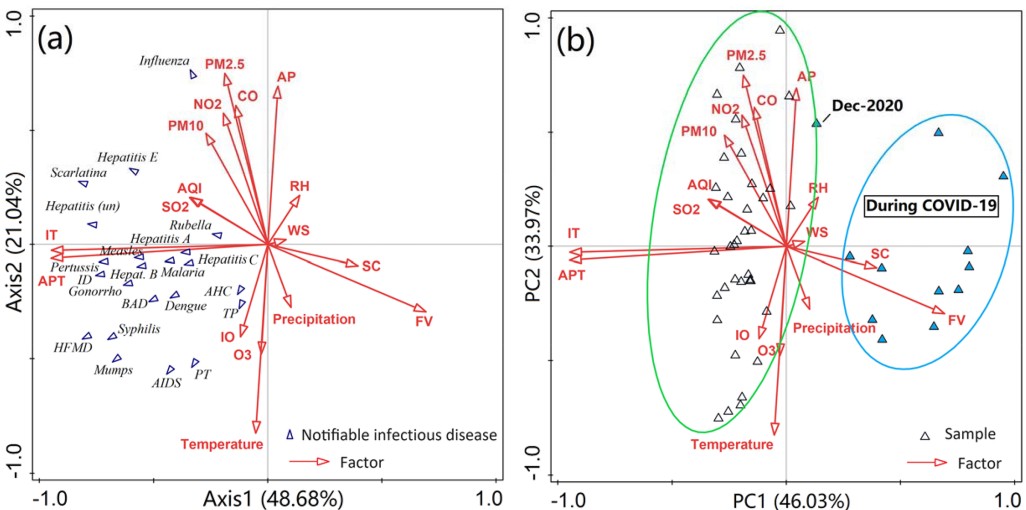

**Figure 3** (A) RDA ordination biplot with environmental variables (arrows) and the most abundant species (notifiable diseases). (B) PCA showing the relationships between environmental variables and samples (month). ID, Infectious diarrhea; AHC, acute hemorrhagic conjunctivitis; BAD, bacillary and amoebic dysentery; HFMD, hand-foot-and-mouth disease; AIDS, acquired immune deficiency syndrome; PT, pulmonary tuberculosis; TP, typhoid and paratyphoid. AP, atmosphere pressure, kPa; Precipitation, mm; d; RH, relative humidity, %; temperature, °C; WS, wind speed, m s$^{-1}$; AQI, air quality index; SC, social consumption, billion yuan; IO, industrial output value, billion yuan; FV, freight volume, million ton; IT, inbound travel, million travelers ; APT, airport passenger throughput, million passengers.

notifiable diseases with different routes of transmission. Among these influencing factors, inbound travel, airport passenger throughput, atmospheric pressure, temperature and PM$_{2.5}$ appeared to be the major factors influencing notifiable diseases, which collectively explained 62.6% of the variance. Correlation analysis showed that inbound travel and airport passenger throughput were positively associated with most diseases, and atmospheric pressure, temperature and PM$_{2.5}$ were closely related with more than half of all diseases (Table S4).

Atmospheric pressure, temperature, and PMs, as well as CO, and NO$_2$ showed a strong correlation with the first axis of the redundancy analysis, each having coefficients exceeding 0.5. The number of inbound visitors and airport passenger throughput were strongly correlated with the second axis. In addition, the PCA analysis also revealed relationships among notifiable diseases, various factors, and samples across two periods (Figs. 2A and 2B). The data on diseases and various factors observed during these periods were clearly grouped into two clusters (December, 2020 excepted). All notifiable diseases were grouped to the left of axis 1, indicating the period before the COVID-19 outbreak. This is in line with the fact that a large number of confirmed cases of notifiable diseases were reported from 2017 to 2019. Similarly, the concentrations of most air pollutants were higher before the COVID-19 pandemic compared to those in 2020, aligning with the previous analysis. These findings also indicate that the COVID-19 pandemic reduced air pollution and had a significant effect on people's daily lives and social environment.

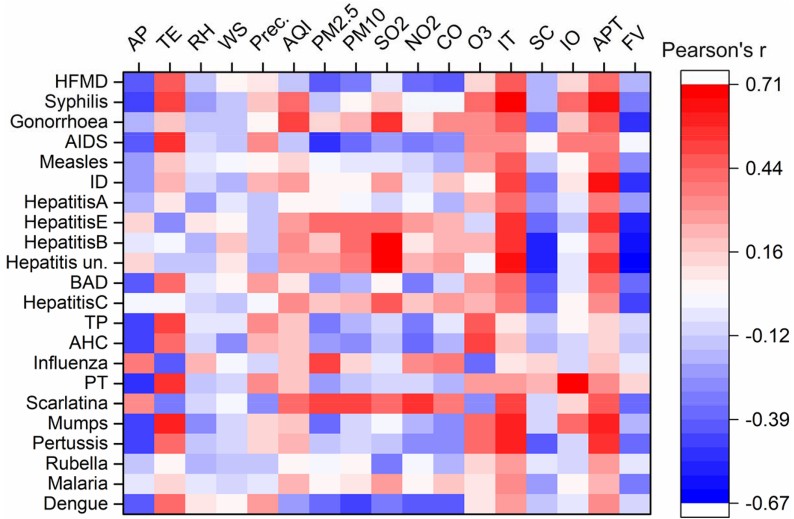

**Figure 4 Correlation analysis between monthly cases of notifiable infectious diseases and monthly averages of various factors.** ID, Infectious diarrhea; AHC, acute hemorrhagic conjunctivitis; BAD, bacillary and amoebic dysentery; HFMD, hand-foot-and-mouth disease; AIDS, acquired immune deficiency syndrome; PT, pulmonary tuberculosis; TP, typhoid and paratyphoid. AP, atmosphere pressure, kPa; Precipitation, mm; d; RH, relative humidity, %; temperature, °C; WS, wind speed, m s$^{-1}$; Prec., precipitation; AQI, air quality index; SC, social consumption, billion yuan; IO, industrial output value, billion yuan; FV, freight volume, million ton; IT, inbound travel, million travelers; APT, airport passenger throughput, million passengers.

Meteorological conditions are closely associated with several infectious diseases. Here, we applied a novel multivariate statistical analysis method to investigate the relationship between notifiable diseases and various variables. For example, as shown in Fig. 4, atmospheric pressure was positively related to influenza and scarlatina, respectively, and negatively correlated with HFMD, syphilis, AIDS, hepatitis E, BAD, TP, PT, mumps, pertussis, and dengue. This is because temperature was strongly inversely associated with atmospheric pressure (Fig. S1B), showing inverse correlations with these diseases compared to atmospheric pressure. However, relative humidity, wind speed, and precipitation did not show significant correlations with these diseases. Modeling studies have shown that influenza and scarlatina are more likely to spread in conditions of low temperature and dry, high-pressure environments (*Duan et al., 2017*; *Jing, Huo & Xiang, 2020*). In contrast, mumps and pertussis have been shown to have a significant positive correlation with increased temperature; for example, in Japan, the weekly number of mumps cases increased by 7.5% for each 1 °C increase in average temperature (*Onozuka & Hashizume, 2011*). As for common infectious diseases such as HFMD, syphilis, gonorrhea and AIDS, the number of cases increased with increased temperature, which may be related to more frequent social activities and enhanced virus activity (*Li et al., 2010*; *Peng et al., 2019*; *Qi et al., 2020*). Most viruses, bacteria and parasites related to infectious disease have a range of critical temperatures, where a deviation too far below or above this range can lead to a decline in fecundity or viability. For instance, Shanghai has a high susceptibility to dengue virus epidemics, transmitted by *Aedes aegypti* mosquitoes, due to

its large seasonal temperature variation offsetting moderate annual average temperatures (25–35 °C) (25–35 °C) (*Huber et al., 2018*). Furthermore, some studies have reported a relationship between infectious diseases and climate changes over long temporal scales. Climate change can alter aqueous and atmospheric environments and microbial growth, and thus further impact infectious disease epidemics (*Karvonen et al., 2010*). These conclusions may provide further evidence for the influence of meteorological factors on common infectious diseases.

Many effective prevention measures were suggested and enforced during the COVID-19 pandemic, providing an opportunity and a basis for scholars to study the influences of the COVID-19 pandemic on air quality (*Mahato, Pal & Ghosh, 2020*; *Tobias et al., 2020*; *Niu et al., 2021*). However, the impact of improved air quality on other infectious diseases received comparatively less attention. In this study, PMs emerged as the primary factors influencing the cases of notifiable diseases. Notably, for several common notifiable diseases, the concentration of $PM_{2.5}$ was significantly positively related to influenza, scarlatina, hepatitis B and D, while being negatively correlated with HFMD, AIDS, BAD, mumps and dengue. Previous studies have indicated that increased concentrations of $PM_{2.5}$ can enhance the transmission of influenza and scarlatina (*Liang et al., 2014*; *Cheng et al., 2020*). Viral hepatitis E and (unspecified) hepatitis were significantly related to PMs and $SO_2$, potentially due to these diseases often breaking out in winter and the following spring (Fig. S3). Meanwhile, dengue, a vector-borne transmitted disease, is endemic during the summer and autumn in Shanghai (*Huber et al., 2018*; *Leal Filho et al., 2018*; *Li et al., 2021*). Air pollutants tend to condense in conditions of stable stratification with low temperatures and high pressure (*Liu et al., 2019a*, *2019b*). Therefore, changes in air pollution may directly affect some airborne-transmitted infectious diseases, rather than all notifiable diseases. During COVID-19, we were able to clearly observe the effect of improved air quality and the consequent decrease in cases of infectious diseases. In other words, the reduction of air pollutants through sustainable policies and new technologies can be a valuable intervention to improve air quality while simultaneously reducing the negative effects of infectious diseases in society (*Qi et al., 2020*).

Population mobility and migration have been verified by several modeling studies as factors that can significantly increase the risk of infectious disease pandemics (*Gushulak & MacPherson, 2000*; *Andrews et al., 2012*; *Felix, Juan & Hecht, 2013*; *Ye et al., 2021*). Population mobility plays a fundamental role in the spatial spread of infectious diseases, regardless of how the diseases are transmitted (*Andrews et al., 2012*; *Poletto, Tizzoni & Colizza, 2013*). In this study, we selected five socioeconomic indicators to analyze their relationships with notifiable diseases. Among these factors, inbound travel and airport passenger throughput were the main factors influencing the spread of nearly all notifiable diseases, with the exception of AHC (Fig. 4). The findings indicate that population mobility, especially inter-regional mobility, enhances transmission. This conclusion is further supported by the results of our correlation analysis (Table S4). The NDOI was significantly related to two indicators of population mobility, suggesting that these indicators contribute more significantly to the spread of notifiable diseases than other factors (Fig. 5). Total retail sales of consumables and flight volume were negatively related

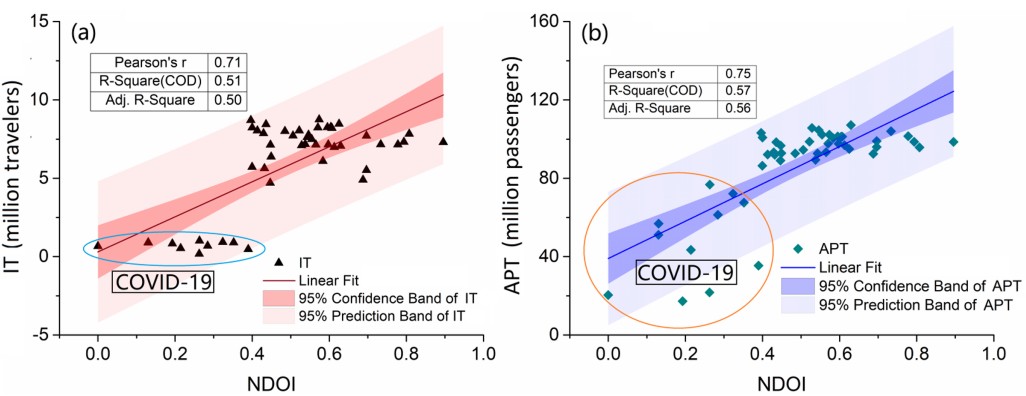

**Figure 5 Correlation analysis between notifiable diseases and indicators of IT (A) and APT (B).** IT, Inbound travel, million travelers; APT, airport passenger throughput, million passengers. The data points on oval represent they were collected during the COVID-19 pandemic.

to some diseases transmitted by water and food, possibly due to rising prices and the transportation demand for goods (Fig. 3). Many predictive models based on population mobility have been established and have served as important references for decision-making. For example, a model assessing the risks of population movement on cross-border transmission of dengue fever highlighted the importance of prioritizing population movements over direct intervention (*Smith & Whittaker, 2014*). The current social networks method has been used to model network effects in influenza spread, considering the travel of individuals and socioeconomic factors in the modeling of influenza spread patterns (*Burris et al., 2021*). Furthermore, the malaria pandemic in Greece during 2003–2012 was a result of travel, immigration to Greece, and visits to friends and relatives (*Maltezou et al., 2013*). In addition, the spread of SARS in 2003, the H1N1 outbreak in 2009, the COVID-19 outbreak in 2019, and recurrent outbreaks of influenza were all closely related to human interactions and mobility (*Bell & World Health Organization Working Group on Prevention of International and Community Transmission of SARS, 2004*; *Balcan et al., 2009*; *Donaldson et al., 2009*; *Ali & Alharbi, 2020*; *Yang, 2020*; *Zhang et al., 2021*).

Decreased disease incidences could be partly driven by altered healthcare-seeking behaviors and surveillance capacity by the pandemic, an inevitable consequence of human-mobility restrictions and overstressed healthcare facilities, which may produce a surveillance biases. In the process of data preprocessing, logarithmic transformations can linearize the data, which may decrease the biases. Therefore, in today's era of globalization, population mobility poses new challenges for infectious disease control. Model development should aim to combine classical models with big data analysis to understand collective behaviors and population mobility on both global and community scales.

## CONCLUSIONS

In this study, we assessed changes in air pollution, meteorological, and socioeconomic factors caused by the COVID-19 pandemic and their influence on the cases of other

notifiable diseases in Shanghai, China, using multivariate statistical analysis. The results revealed that the COVID-19 pandemic improved air quality and led to significant socioeconomic changes. During the COVID-19 pandemic, the concentrations of $PM_{10}$, $SO_2$, $PM_{2.5}$, $NO_2$, and AQI decreased by 18.0%, 30.4%, 11.5%, 10.6%, and 10.0%, respectively. In addition, the COVID-19 outbreak also influenced socioeconomic indicators. Compared to the period before the COVID-19 pandemic, the number of monthly inbound travelers and airport passenger throughput decreased by 85.5%, and 47.4%, respectively. Moreover, the total cases and NDOIs of all notifiable diseases decreased by 47.1% and 52.6%, respectively, compared to before the pandemic. This suggests that the NDOI could be effectively used as an integrated index to reflect the relative intensity and fluctuations of notifiable diseases. Our results also showed different responses of various notifiable diseases to the three categories of indicators. Factors such as population mobility, including inbound travelers and airport passenger throughput, along with $PM_{2.5}$, atmospheric pressure, and temperature, were the main factors influencing the spread of notifiable diseases. While meteorological factors may be beyond our control, stringent prevention policies addressing population mobility and air pollutant emissions may have a positive long-term impact on containing the spread of infectious diseases. Future studies should aim to combine classical modeling with big data analysis for forecasting a composite index of notifiable diseases on different scales.

### Funding

This work was supported by the Key R&D and Promotion Projects of Henan Province (No. 232102320137), the Key Scientific Research Projects of Universities and Colleges in Henan Province, China (No. 23B610010) and the Scientific Research Foundation for High-level Professionals of Zhoukou Normal University (No. ZKUNC2019006). The funders had no role in study design, data collection and analysis, decision to publish, or preparation of the manuscript.

### Grant Disclosures

The following grant information was disclosed by the authors:
Key R & D and Promotion Projects of Henan Province: 232102320137.
Key Scientific Research Projects of Universities and Colleges in Henan Province, China: 23B610010.
Scientific Research Foundation for High-level Professionals of Zhoukou Normal University: ZKUNC2019006.

### Competing Interests

The authors declare that they have no competing interests.

### Author Contributions

- Yongfang Zhang conceived and designed the experiments, performed the experiments, authored or reviewed drafts of the article, and approved the final draft.

- Wenli Feng conceived and designed the experiments, analyzed the data, prepared figures and/or tables, and approved the final draft.

## Data Availability

The raw data are available in the Supplemental File.

## Supplemental Information

Supplemental information for this article can be found online at http://dx.doi.org/10.7717/peerj.17124#supplemental-information.

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
