# Peer review of "Impact of the coronavirus disease 2019 pandemic on the diversity of notifiable infectious diseases: a case study in Shanghai, China"

_PeerJ, doi:10.7717/peerj.17124_

## Round 0.1 · original submission · Major Revisions

Dear Drs. Feng and Zhang:

Thanks for submitting your manuscript to PeerJ. I have now received three independent reviews of your work, and one reviewer recommended rejection, while the others suggested minor revisions. I am affording you the option of revising your manuscript according to all three reviews but understand that your resubmission may be sent to at least one new reviewer for a fresh assessment (unless the reviewer recommending rejection is willing to re-review).

In general, the reviewers wish to see improvements to English and grammar, as well as a better presentation of your findings (emphasizing the relevance of your work to the field). The methods should be clear, concise and repeatable. Please ensure this, and make sure all relevant information and references are provided.

There are many suggestions to improve the manuscript.

Good luck with your revision,

Best,

joe

**Language Note:** The review process has identified that the English language must be improved. PeerJ can provide language editing services - please contact us at copyediting@peerj.com for pricing (be sure to provide your manuscript number and title). Alternatively, you should make your own arrangements to improve the language quality and provide details in your response letter. – PeerJ Staff

Reviewer 1 ·

Basic reporting

1) Unless there is a specific rationale, it is advisable to separate the Results and Discussion sections, following the conventions of professional scientific articles. (Use these Standard Sections where possible. https://peerj.com/about/author-instructions/#standard-sections)The integration of results and discussion can lead to ambiguity in discerning the outcomes of this study.
2) It is essential to include statements regarding the limitations of the study.
3) In the Introduction and Discussion sections, there is a need for improved organization to streamline the flow of both sections. This can be achieved by omitting or relocating detailed data from the Introduction to the Discussion section, thus enhancing conciseness. Additionally, the Introduction should encompass crucial contextual information, such as the timing of the initial detection of COVID-19 cases in Shanghai and the timing and specifics of the implemented measures, for interpreting the monthly data on the COVID-19 pandemic within the study area.
4) To enhance clarity and coherence, it is suggested that the order of objectives align with the order of results.
5) The manuscript contains several typographical errors (e.g., line 219 HMFD, line 250 Fig2?), necessitating thorough English proofreading. Additionally, abbreviations should be spelled out the first time they are introduced (e.g., line 106 AQI).

Experimental design

1)While the research questions are clearly articulated, the identification of the specific knowledge gap is not evident, despite a previous report from China addressing a similar issue (Geng, M. J., et al. Nat Commun. 2021;12(1):6923).
2)Line 96 (c) explore the critical factors influencing the NIDs.
Line 97-98 The findings will be helpful to forecast the situation of infectious disease epidemic
it's important to note that this study employs an ecological design, making it challenging to infer causal relationships. Therefore, the above sentences may not be appropriate.
3) If you are discussing aggregated case numbers, it is imperative to clarify in the methods section that the data are derived from all notifiable disease surveillance. If the aggregated case counts include cases from sentinel surveillance, they may not be appropriate for this purpose.
4)Line 212-213 The correlation between NDOI and the number of disease case or type were significant, which means the number of disease case or type can determine NDOI (Figs. 2b and 2c).
Correlation tests are not predictive of one variable's value based on another, if my understanding is correct. Clarification on this point can be found here(https://www.bmj.com/about-bmj/resources-readers/publications/statistics-square-one/11-correlation-and-regression)。

Validity of the findings

1) Line 156-157 Therefore, the COVID-19 pandemic has no effect on metrological factors.
Line272-273 Our results revealed atmospheric pressure and temperature were key meteorological factors influencing notifiable diseases.
It's essential to recognize that this study employs an ecological design, making it challenging to establish causal relationships from these findings. Therefore, the above sentences may not be appropriate.
2) Regarding the section on the Influence of COVID-19 on notifiable diseases, it's worth noting that Syphilis, Gonorrhea, and AIDS are primarily sexually transmitted diseases, while Measles is an airborne disease. Analyzing them collectively as Direct-contact transmitted diseases may not be appropriate. Additionally, the category of Airborne transmitted diseases includes both Airborne transmitted diseases and Droplet transmitted diseases, which may not be a suitable classification.
3)In the discussion, it's important to consider surveillance biases that may have been particularly relevant during the COVID-19 pandemic.
Decreased disease incidences could be partly driven by altered healthcare-seeking behaviors and surveillance capacity by the pandemic, an inevitable consequence of human-mobility restrictions and overstressed healthcare facilities. Such bias could be more severe in the early wave (e.g., Phase II) of the pandemic (Geng, M. J., et al. Nat Commun. 2021;12(1):6923.).

Additional comments

1)Given the variability in the occurrence of COVID-19, the implementation of infection control measures, and the selection of notifiable diseases across different countries or time periods, we recommend that the title incorporate information on time and location.
2) Overall, the terms outbreak, epidemic, and pandemic need to be used properly(e.g. line 162)(https://www.cdc.gov/scienceambassador/nerdacademy/defining-the-pandemic.html).
3) Line 229-230 Malaria 230 and dengue are viral infection transmitted by mosquitoes,
Malaria is a parasitic disease, not a viral one.
4)Line 170 COVID-19→COVID-19 pandemic?

Reviewer 2 ·

Basic reporting

In examining the repercussions of the COVID-19 pandemic on the diversity of notifiable infectious diseases, the authors provide a comprehensive report. They innovate by introducing a metric, the Index of Notifiable Disease Outbreak (NDOI), which serves to quantify COVID-19's impact on such diseases. Their research notably underscores three crucial determinants for the spread of these diseases: population mobility, atmospheric pressure, and temperature.

While the manuscript largely flows with clarity and coherence, there are areas where redundancy could be minimized for enhanced clarity. A case in point is line 165, where the authors observe an increase in "freight volume and total retail sales of consumer goods by 32.9% and 25.8%" respectively. This statement resurfaces in line 194, accompanied by a rationale attributing the surge to "tight supplies and a heightened demand for social consumer goods." For the sake of clarity and succinctness, the explanation should accompany the observation upon its first mention.

Furthermore, the authors excel in grounding their findings within the existing scholarly discourse. They seamlessly weave relevant literature throughout, ensuring that their insights are contextualized and that all referenced works are appropriately acknowledged.

Experimental design

The methods section is articulately presented. The precise formula for the notifiable disease outbreak index is delineated with clarity. Furthermore, the section detailing the statistical analysis is comprehensive, facilitating readers' understanding of the study's structure and enabling them to replicate the results. The authors' use of an array of statistical methods to examine the correlation between COVID-19 and notifiable infectious diseases significantly strengthens the paper.

Validity of the findings

The figures in this manuscript effectively illustrate and bolster the primary claims and findings. However, certain areas require revision for clarity:
- In Figure 3, both hollow and solid triangles are used, but the legend lacks explanations for these symbols, potentially leading to confusion among readers.
- For Figure 4, the z title in the heatmap and its corresponding legend is not defined, necessitating clarification.

Additionally, there are referencing discrepancies:
- In line 250, the reference should be to Figure 3, not Figure 2.
- In lines 262-263, the correct references should be Figure 3a and 3b, rather than Figure 2a and 2b.

Lastly, there are specific numerical values mentioned between lines 215-233 that appear to be missing from Table 2. A thorough review to verify these figures and enhance the clarity of their references is advised.

Reviewer 3 ·

Basic reporting

1. Line 197: suggest rephrasing “Of course, the socioeconomic response...”

2.Line 91-93: suggest revising the sentence “Therefore, it is interesting and worth studying the response mechanism of other NIDs with various transmission routes to the changes in environmental and socioeconomic factors caused by COVID-19 pandemic.”

3. Figure 1, 2, 4, and 5 could significantly enhance their impact with the inclusion of additional details:
(1)Figure 1: It would be beneficial to provide a clear title directly on the figure and offer individual general descriptions in the figure legend for each of the sub-figures.
(2)Figure 2: It is advisable to specify the type of analysis employed to generate these figures to enhance clarity.
(3)Figure 4: In the figure legend, please include the figure type (heat map) and specify the data source.
(4)Figure 5: Clarify the meaning of the data represented by the circles (blue in 1, red in 2). Additionally, consider adding a title to the figure for improved context and comprehension.

4. The tables in the manuscript are well-organized, but due to the volume of data, they can appear crowded, making it challenging to extract key information. To enhance readability and comprehension, the following recommendations are suggested:
(1)Table 1: Consider presenting the information in Table 1 as bar graphs. You can create multiple bar graphs, each focusing on specific classifications or parameters. Alternatively, you can choose a few example cities and create bar graphs that illustrate the data for each parameter, such as O3, NO2, etc.
(2)Table 2: Transform Table 2 into a line graph to illustrate the trend of NDOI decreasing over time. Additionally, consider creating a reverse bar graph to depict the rate of change.
By implementing these graphical representations, the presentation of data will become more straightforward and accessible for readers, facilitating a clearer understanding of the key findings.

Experimental design

1. The study predominantly relies on data from June 2016 to May 2021. Consequently, there is a possibility of biases in disease reporting between the pre-pandemic and pandemic periods, which could potentially undermine the study's outcomes. To enhance the robustness and credibility of the model and analysis, it is advisable to perform a sensitivity analysis to assess the resilience of the results in the face of such potential biases.

Validity of the findings

1. Paragraph Line 234-247 focus on the impact of protective measures implemented during the COVID-19 pandemic, such as mask-wearing, social distancing, and hygiene practices, on the reduction of cases of other notifiable infectious diseases. While the paragraph suggests a decrease in cases of other infectious diseases, it is important to acknowledge that the available data provides evidence for the reduction in cases but falls short in establishing a direct causal link between these protective measures and the decline in disease cases. To strengthen the argument and support the claim that protective measures can indeed reduce cases of other diseases, it is essential to incorporate additional data that can establish a causal mechanism between the implementation of these measures and the observed reduction in disease cases. This will provide a more comprehensive and convincing basis for the conclusions drawn in this paragraph.

---

## Round 0.2 · Minor Revisions

Dear Drs. Zhang and Feng:

Thanks for revising your manuscript. The two reviewers willing to re-review are mostly satisfied with your revision (as am I). Great! However, there are some additional issues to entertain. Please address these ASAP so we may move forward with your manuscript.

Importantly, please provide all relevant information in your figures, making sure they are stand-alone (with their cognate legends). Please ensure that your workflow is repeatable.

Good luck with your revision,

Best,

-joe

Reviewer 2 ·

Basic reporting

The authors addressed all my comments.

Experimental design

The authors addressed all my comments.

Validity of the findings

The authors addressed the majority of my comments. However, note that the explanation for the symbols in Figure 3 and the name for the z title in Figure 4 are not added in the current review proof.

Reviewer 3 ·

Basic reporting

The issues with phrasing previously identified as 1 and 2 have been effectively addressed. The current sentence structure is clear and accurate.

While the rebuttal letter provides thorough explanations for the changes made to figures 1, 2, 4, and 5, it appears that the figure descriptions or legends in the revised manuscript are incomplete. Specifically for example, in the legend for figure 2, the sentence referenced in the rebuttal letter concludes abruptly at "NDOI and the." It is important to ensure that the entire legend text is included in the manuscript. Despite this, all the revisions mentioned in the rebuttal letter are commendable and well-executed.

Regarding the organization of tables 1 and 2, the preference for bar graphs mentioned in previous comments still stands. However, the author's rationale for continuing with tables is understandable. The choice to use tables is acceptable.

Experimental design

The newly conducted resilience analysis is well-crafted and executed. It successfully addresses the previous flaw that was identified.

Validity of the findings

The additional data and details provided in lines 239-262 effectively address the requirement for establishing a causal mechanism between the implementation of protective measures and the observed decline in disease cases.

---

## Round 0.3 · accepted · Accept

Dear Drs. Feng and Zhang:

Thanks for revising your manuscript based on the concerns that were raised. I now believe that your manuscript is suitable for publication. Congratulations! I look forward to seeing this work in print, and I anticipate it being an important resource for groups studying the impact of COVID-19 on other infectious diseases. Thanks again for choosing PeerJ to publish such important work.

Best,

-joe